# Association Analysis of the Cerebral Fractional Tissue Oxygen Extraction (cFTOE) and the Cerebral Oxygen Saturation (crSaO_2_) with Perinatal Factors in Preterm Neonates: A Single Centre Study

**DOI:** 10.3390/jcm11123546

**Published:** 2022-06-20

**Authors:** Melinda Matyas, Mihaela Iancu, Monica Hasmasanu, Anca Man, Gabriela Zaharie

**Affiliations:** 1Neonatology Department, “Iuliu Hațieganu” University of Medicine and Pharmacy, 3, Clinicilor Street, 400012 Cluj-Napoca, Romania; melinda.matyas@umfcluj.ro (M.M.); popa.monica@elearn.umfcluj.ro (M.H.); gzaharie@umfcluj.ro (G.Z.); 2Department of Medical Informatics and Biostatistics, “Iuliu Hațieganu” University of Medicine and Pharmacy, 8, Victor Babes Street, 400012 Cluj-Napoca, Romania; 3Neonatology Department, County Emergency Hospital, 3, Clinicilor Street, 400347 Cluj-Napoca, Romania; mary_a_90@yahoo.com

**Keywords:** cerebral oxygen saturation (crSaO_2_), near infrared spectroscopy (NIRS), premature newborns

## Abstract

(1) Background: Near-infrared spectroscopy (NIRS) is a non-invasive, easily performed method of monitoring brain oxygenation. The regional cerebral oxygen saturation (crSaO_2_) and the cerebral fractional tissue oxygen extraction (cFTOE) evaluated by NIRS provide more accurate information on brain oxygenation than the blood oxygen saturation. We investigated the effect of perinatal factors on cerebral oxygenation of preterm newborns. (2) Methods: We conducted a longitudinal study with 48 preterm newborns <34 weeks of gestation who underwent NIRS registration during the first 72 h of life. crSaO_2_ was measured and cFTOE was calculated foreach patient. (3) Results: One-way ANOVA showed no significant main effect of IVH severity on crSaO_2_ and cFTOE (*p* > 0.05); there was a tendency toward statistical significance concerning the difference between the means of crSaO_2_ (*p* = 0.083) and cFTOE (*p* = 0.098). Patients with intraventricular haemorrhage (IVH) had a lower mean of crSaO_2_ and a higher mean of cFTOE (59.67 ± 10.37% vs. 64.92 ± 10.16% for crSaO_2_; 0.37 ± 0.11 vs. 0.32 ± 0.11 for cFTOE) compared to those with no IVH. Significantly lower values of crSaO_2_ and higher values of cFTOE were found in neonates receiving inotropic treatment (*p* < 0.0001). Episodes of apnoea also proved to influence the cerebral oxygen saturation of the study group (*p* = 0.0026). No significant association between the maternal hypertension treatment and the cerebral oxygenation of preterms was found. (4) Conclusions: This study showed a decreased cerebral oxygen saturation of preterms with IVH, inotrope support and apnoea episodes.

## 1. Introduction

The neurological evolution of preterm infants is influenced by multiple perinatal factors. Among the prenatal factors, maternal inflammation, chorioamnionitis or pre-eclampsia influence the neurological evolution [1,2]. Immediately after birth, several conditions characteristic of prematurity, and nutrition, caring and nursing have an impact on the neurological development of the premature infant. After clamping the umbilical cord, the newborn must maintain adequate oxygenation and the flow of nutrients to the brain using incompletely known self-regulatory mechanisms [1,3,4].

Monitoring the blood oxygen saturation or the heart rate does not accurately reflect the oxygenation in the brain. By using near-infrared spectroscopy (NIRS), a non-invasive technique, we can determine the regional cerebral oxygen saturation (crSaO_2_) and the cerebral fractional tissue oxygen extraction (cFTOE) in the perinatal period. Oxygen is an important substrate for brain metabolism. Hyperoxia and hypoxia are extremely harmful to the brain, especially in premature newborns, who due to the immaturity of their central nervous system are much more vulnerable to oxygen saturation than full-term newborns [2,5].

This measure of crSaO_2_ reflects the oxygen saturation in a mixed vascular system dominated by venules. Fractional tissue oxygen extraction (cFTOE) is calculated based on crSaO_2_ and transcutaneous arterial oxygen saturation (tcSaO_2_) values [1,2,3,4]. crSaO_2_ serves as a marker of cerebral hypoxia. cFTOE reflects the balance between the cerebral oxygen delivery (cerebral perfusion) and the cerebral oxygen consumption and thus serves as an indicator of cerebral ischemic hypoxia [2,4,5,6].

The NIRS is a more accurate indicator of brain oxygenation than monitoring the blood oxygen saturation, allowing the oxygen supply to be adjusted and therapeutic corrections (which cannot be performed based on blood saturation of oxygen) to be made (e.g., initiating inotropic support) [3,6].

The aim of this study was to evaluate the brain tissue oxygen extraction (cFTOE) and the regional brain saturation (crSaO_2_) in premature infants. We investigated the associations of maternal factors with these parameters and the role of perinatal disorders that cause changes in cFTOE and crSaO_2_.

## 2. Materials and Methods

### 2.1. Design

We conducted a longitudinal study in the neonatology department of the Clinic of Obstetrics and Gynecology I of the County Clinical Emergency Hospital, Cluj-Napoca, Romania. In the study, preterm newborns with a gestational age less than 34 weeks + 6 days and admitted between January 2016–December 2017 in the third-level intensive care unit were enrolled.

Brain oxygenation, oxygen saturation and pH gas values were monitored in all newborns of the study group during the first 3 days of life. All patients included in the study were also evaluated by point of care ultrasound for the diagnosis and severity of cerebral haemorrhage.

The study was approved by the Institutional Review Board of the County Emergency Hospital, Cluj-Napoca, Romania.

### 2.2. Analysed Parameters

#### 2.2.1. Brain Oxygenation

The brain oxygenation was monitored during the first 3 days of life with the INVOS 4100 (Somanetics Corporation, Covidien, Watford, UK, Troy) using the Neonatal Oxyalert NIRS sensor to measure crSaO_2_ values. The optical sensor measures the quantity of reflected light photons as a function of 2 wavelengths (730 and 805 nm) and determines the spectral absorption of the underlying tissue. Because the oxygenated haemoglobin and the deoxygenated haemoglobin have distinct absorption spectra, NIRS can differentiate between the two. The ratio of oxygenated haemoglobin to total haemoglobin reflects the regional oxygen saturation of cerebral tissue. crSaO_2_ was measured in the first 72 h of life by continuous monitoring. The optical sensor was placed on the left frontoparietal side of the infant’s head and held in place with an elastic bandage. The sensor was placed in the first hour of life of the newborn. We calculated cFTOE as cFTOE = (tcSaO_2_ − crSaO_2_)/tcSaO_2_. cFTOE reflects the balance between the cerebral oxygen supply and the cerebral oxygen consumption. cFTOE reflects hypoxic distress better than crSaO_2_ [7].

#### 2.2.2. Oxygen Saturation and Blood Pressure

Simultaneously, we measured tcSaO_2_ by pulse oximetry, using a Mindray monitor. In parallel, we monitored the blood pressure, respiratory rate and the mean arterial blood pressure (MAP). The values were recorded every two hours. Based on the values recorded for blood pressure and respiratory rate, we were able to calculate the average value of these parameters.

#### 2.2.3. Head Ultrasound

A head ultrasound was performed on all newborns in the study group in the first 72 h of life. The ultrasound of the hospital ward was used with the 8 Hz probe.

#### 2.2.4. pH Gas Value

pH gas value parameters were monitored by analysing the venous blood, according to the evolution of the tcSAO_2_, of the clinical outcomes and the changes made to the ventilation parameters.

#### 2.2.5. Clinical Variables

Prospectively, we collected details on perinatal and neonatal characteristics that might influence the hemodynamic. These included the gestational age, birth weight, Apgar score, birth asphyxia, early-onset signs of circulatory failure and inotrope medication, ventilatory status, patency of the ductus arteriosus and medication. Maternal and pregnancy-related variables such as medication, pre-eclampsia and signs of maternal intrauterine infection were collected. Maternal pre-eclampsia was defined as systolic blood pressure ≥160 mmHg and diastolic ≥90 mmHg associated with proteinuria [2,8].

The presence and severity of respiratory distress and the surfactant therapy were assessed. Their correlation with brain oxygenation was examined. The diagnosis of respiratory distress was established based on clinical criteria (chest retraction, thoraco-abdominal balance, nasal flaring, grunting and radiological findings (namely reticulogranular patterns, air bronchograms and ground glass opacity). Depending on the clinical symptoms, 3 forms of severity of respiratory distress were established. Surfactant therapy for severe cases of distress was administered based on the recommendations of the European guideline [9]. The presence of apnoea in the preterm infants in the study was quantified. Apnoea episodes were defined as decreases in tcSaO_2_ accompanied by bradycardia, which required at least tactile stimulation. Caffeine therapy in the first 2 h of life was initiated in all newborns in the study group to prevent apnoea attacks.

The diagnosis of early sepsis was established based on clinical criteria associated with paraclinical changes such as changes in blood count, the presence of inflammatory syndrome and possibly a positive blood culture. The persistence of ductus arteriosus was established by cardiac ultrasound performed in the first 72 h. Cerebral haemorrhage was diagnosed by head ultrasound and was classified in 4 degrees of severity based on imaging criteria [10]. For each patient, we measured the resistivity index (IR) at a median cerebral artery level. The ultrasound was carried out with a GE (General Electric) device, in two sections—sagittal and coronal—by a neonatologist experienced in head ultrasound using the 8 Hz probe. Asphyxia in the study group has been diagnosed based on an Apgar score ≤ 3 for ≥10 min, pH ≤ 7, in the first hour of life or base deficit ≥ 16 mmol/L [11].

### 2.3. Statistical Analysis

Demographic and clinical characteristics of premature neonates were summarised by absolute and relative frequencies (%); arithmetic mean ± standard deviation; or median with interquartile interval, IQR = [Q1, Q3], where Q1 = first quartile and Q3 = third quartile.

One of the studied parameters was the fractional tissue oxygen extraction (cFTOE), which was calculated based on the mean of crSaO_2_ and on the mean of the transcutaneous arterial oxygen saturation (tcSaO_2_) values, as follows: cFTOE = (mean of crSaO_2_ − mean of tcSaO_2_)/mean of tcSaO_2_.

The distributions of continuous characteristics (average crSaO_2_, cFTOE, mean of arterial blood pressure (MAP), IR, tcSaO_2_, and tcSaO_2__FiO_2_) were checked for univariate normality using different methods (Shapiro–Wilk test, normal Q-Q plot, skewness–kurtosis graph).

The association between the cerebral fractional tissue oxygen extraction (cFTOE) and the cerebral oxygen saturation (crSaO_2_) with perinatal characteristics was assessed using the Student-*t* test for independent samples or one-way ANOVA.

The correlation between the cerebral fractional tissue oxygen extraction (cFTOE) and the cerebral oxygen saturation (crSaO_2_) with IR, MAP, tcSatO_2_ and tcSatO_2__FiO_2_ was investigated using Spearman’s rank correlation coefficient (ρ), considering that distributions of continuous clinical perinatal characteristics did not follow a Gaussian distribution. To quantify the effective size of association, a 95% confidence interval (CI) for ρ was calculated using the percentile method based on 1000 bootstrapped samples with replacement. The lower and upper limits of 95% CI were the 2.5th and 97.5th percentiles.

For all two-sided statistical tests, a *p*-value lower than the significance level, α = 0.05, was considered as a significant result. All statistical analyses were performed using R software version 4.1.1 (R Foundation for Statistical Computing, Vienna, Austria).

## 3. Results

### 3.1. Description of Preterm Neonates Sample

We analysed a sample of 48 premature neonates with the mean gestational age 27.98 ± 2.37 weeks and median birth weight 1035 g (IQR 740–1270 g); ten patients were outborn, and two neonates died. The demographic and clinical characteristics of the study sample are presented in Table 1.

### 3.2. Relationship between cFTOE, crSaO_2_ and Maternal Pathology

Nifedipine was administered to 16 (33.33%) pregnant women. We found no significant difference in the mean crSaO_2_ between the preterm neonates whose mothers received hypertension treatment and those whose mothers did not receive therapy (Student-*t* test, *p* = 0.263). Similar results were obtained regarding the means of cFTOE (Student-*t* test, *p* = 0.245).

### 3.3. Relationship between Cerebral Fractional Tissue Oxygen Extraction (cFTOE) and Cerebral Oxygen Saturation (crSaO_2_) with Demographic and Clinical Characteristics of Premature Neonates: Bivariate Analysis

We found no significant difference in the mean values of crSaO_2_ (*p* = 0.212) or cFTOE (*p* = 0.282) between boys and girls (Table 2). There was no significant main effect of IVH severity on crSaO_2_ and cFTOE (one-way ANOVA *p* > 0.05); however, patients with IVH had a lower mean of crSaO_2_ and a higher mean of cFTOE: mean ± standard deviation: 59.67 ± 10.37% vs. 64.92 ± 10.16% for crSaO_2_ (*p* = 0.083); 0.37 ± 0.11 vs. 0.32 ± 0.11 for cFTOE (*p* = 0.098)

We found no significant main effect of RDS severity on crSaO_2_ (*p* = 0.372) and cFTOE (*p* = 0.449), but there was a tendency toward significance concerning difference in crSaO_2_ (*p* = 0.076) in newborns with respiratory distress who needed surfactant therapy compared to those whose respiratory distress did not require this therapy (Table 2).

As seen in Table 2, there was a highly significant association between apnoea and crSaO_2_ (*p* = 0.0026), as well as between apnoea and cFTOE (*p* = 0.0039). Furthermore, a lower crSaO_2_ and a higher cFTOE were found in patients receiving inotropes for low blood pressure (*p* < 0.0001). Hypocapnia (identified in at least two measurements of pH) was associated with significantly higher values of crSaO_2_ compared to normocapnia (65.04 ± 9.06% vs. 59.05 ± 11.32%, *p* = 0.047).

There was a significant correlation of crSaO_2_ and cFTOE with MAP in Day 1–Day 4 (Table 3, Figure 1) but not with IR (*p* > 0.05).

## 4. Discussion

The present study was carried out to evaluate the influence of maternal and perinatal pathologies on crSaO_2_ and FTOE in the premature newborn. Nifedipine therapy was administered to 16 mothers of premature babies. We did not find any significant association of the brain oxygenation indices (crSaO_2_ and FTOE) with this therapy. MAP also did not show changes in preterm infants whose mothers received therapy compared to those whose mothers did not receive Nifedipine. These findings are similar to those of Thewissen et al. They analysed the influence of maternal Labetalol therapy on neonatal brain oxygenation in preterm infants and found no influence of Labetalol on the cerebral oxygenation, heart rate or mean blood pressure in exposed premature infants [12]. The maternal hypertension treatment may influence the cerebral vascular self-regulation. This was not observed in our study. However, the study group included a small number of patients and a small number of cases treated with Nifedipine, which might explain the lack of correlation observed.

Some neonatal complications related to the preterm birth had an impact on brain oxygenation in the study group. All newborns in the study group received caffeine therapy, initiated in the first 2 h of life, and required respiratory support. The presence of apnoea episodes with tcSatO_2_ < 80%, and bradycardia, more than 10 episodes in 24 h, during the first days of life were associated with a decrease in cerebral oxygenation, compared to premature infants, in whom the episodes of desaturation were fewer. This can represent an important element of monitoring for preterm neonates. In addition to tcSatO_2_, the crSaO_2_ and FTOE values allow for direct information on brain oxygenation process during the apnoea and, subsequently, a prompt therapeutic intervention [13,14,15,16,17]. It is important to maintain these parameters in normal range in preterm neonates because low oxygenation is associated with a poor neurodevelopmental outcome [18,19,20].

The effect of bradycardia on brain oxygenation is also described by Walter et al. They showed a much greater impact on brain oxygenation in the very preterm than in the late preterm. Even mild bradycardias are associated with falls in cerebral oxygenation [13,17,21]. In our study, the need for inotropic support (established based on the blood pressure values) was correlated with cerebral oxygenation: in the group of preterms receiving inotrope medication in the first 3 days of life, the crSaO_2_ was significantly lower (*p* < 0.0001) than at untreated ones. A low MAP value also was corelated with a poor cerebral oxygenation.

For patients included in this study, a head ultrasound was performed within the first 72 h of life. As expected, in those with cerebral haemorrhage, brain oxygenation was lower. Brain injury in premature infants appears as a complex association of primary diseases and secondary maturational and trophic disturbances, rather than as the result of a single agent [22,23,24,25]. Previously, relationships between the severity of GMHIVH or PVHI and cerebral oxygenation were found [22,26,27,28]. In these cases, however, cerebral oxygenation was only measured during the first days after birth. Contrary to the findings in these studies, we found no difference in rcSO_2_ r FTOE in the infants with mild haemorrhages compared with the infants with severe haemorrhages. However, measurements on the third day in infants with mild haemorrhages, showed a trend toward lower crSaO_2_ and higher FTOE than in infants without haemorrhages. Still, our findings are limited by the small number of severe IVHs in our study. Vehagen et al. found in their study differences in cerebral oxygenation of preterm newborns with IVH lasting at least for the first 2 weeks of life. An increased FTOE can be linked to a lower oxygen supply or an increased oxygen consumption. A lower cerebral blood flow is associated with low oxygen supply. [26].

Previous studies have shown that the average value of the rScO2 in the preterm neonates was ∼65% at admission. The mean cFTOE ranges from 0.25 to 0.34 during the first 72 h of life in preterm babies [18,29,30,31]. In our study, we found a lower crSaO_2_ in preterm neonates below 28 weeks of gestation than in the ones above this gestational age (59.26 vs. 64.28). The cFTOE had the same trend.

Other neonatal conditions such as respiratory distress syndrome, asphyxia or neonatal early onset sepsis, and low pH (<7.2) did not correlate with crSaO_2_ and FTOE values.

NIRS plays an important role in the assessment of interventions and therapies in neonatal care. Routine NIRS monitoring of cerebral oxygenation in NICUs may increase staff awareness for interventions to reduce the repetitive falls in cerebral oxygenation in preterm infants [7,32,33].

This study showed changes in cerebral oxygen saturation of preterm newborns with IVH or requiring inotropic support and surfactant treatment. Similar findings were described by other authors [34,35,36]. A greater need for mechanical ventilation and lower cerebral oximetry index was found for preterm neonates requiring dopamine treatment for hypotension [35].

The limitations of this study consist of a relatively small number of preterm infants and the single-centre study. The present study should be regarded as exploratory research that investigated the associations of maternal factors and perinatal disorders with cFTOE and crSaO_2_. Considering the small sample size, we could not test a multivariable model containing all relevant clinical factors that can explain variability of SctO_2_/cFTOE. A future study with a higher sample size is necessary to confirm the independent maternal factors or perinatal disorders as potential predictors of SctO_2_/cFTOE changes. Despite the relatively small sample size, our findings are in accordance with other studies finding a sustained role and importance of the cerebral oxygen saturation monitor in the NICU patient.

## 5. Conclusions

In preterm newborns, the cerebral oxygenation was associated with different factors, both maternal and perinatal. Our study showed that the IVH, the need for inotrope support and episodes of apnoea were correlated with crSaO_2_ and cFTOE. There was a trend of decreased crSaO_2_ and cFTOE values with younger gestational age of the neonates. We found no significant association between the maternal hypertension treatment and the cerebral oxygen saturation of the preterm infants.

## Figures and Tables

**Figure 1 jcm-11-03546-f001:**
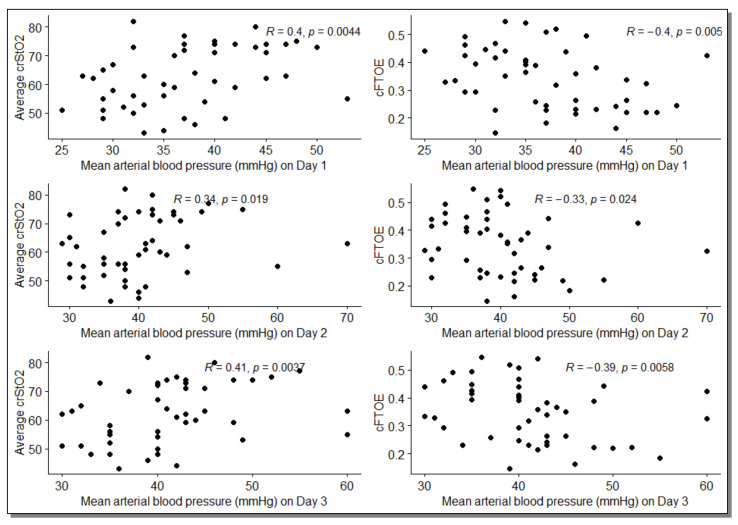
Scatter plots between the mean arterial blood pressures measured in the first 3 days and crSaO_2_ and cFTOE; R = Rho (Spearman’s rank correlation coefficient).

**Table 1 jcm-11-03546-t001:** Characteristics of the premature neonate group.

Variables	Sample Statistics
Gestational Age (weeks)	27.98 ± 2.37
Extremely premature ^(1)^	19 (39.58)
Birth weight (g)	1035 [740, 1270]
Extremely_low_birth weight ^(2)^	23 (47.92)
Gender (male)	26 (54.17)
RDS category	
No/mild	5 (10.42)
Moderate	13 (27.08)
Severe	30 (62.50)
Asphyxia	17 (35.42)
Anaemia	21 (43.75)
Sepsis	24 (50.00)
IVH severity	
Absent	24 (50.00)
I–II	17 (35.42)
III–IV	7 (14.58)

^(1)^ Gestational age < 28 weeks; ^(2)^ birth weight <1000 g; RDS = respiratory distress syndrome, IVH = intraventricular haemorrhage; data were expressed using descriptive statistics as mean ± standard deviation and absolute (relative) frequencies.

**Table 2 jcm-11-03546-t002:** Associations of cFTOE and crSaO_2_ with clinical characteristics of premature neonates.

Variables	Average crStO_2_	*p*-Value	cFTOE	*p*-Value
Sex ^(b)^		0.212		0.282
Male	60.54 ± 9.31		0.36 ± 0.10	
Female	64.36 ± 11.61		0.33 ± 0.12	
Extremely premature		0.106		0.138
Gestational age < 28 weeks	59.26 ± 9.13		0.37 ± 0.09	
Gestational age ≥ 28 weeks	64.28 ± 10.99		0.33 ± 0.14	
Low birth weight		0.631		0.756
Birth weight < 1000 g	61.52 ± 9.76		0.35 ± 0.10	
Birth weight ≥ 1000 g	63.00 ± 11.28		0.34 ± 0.12	
IVH		0.215		0.244
absent	64.92 ± 10.16		0.32 ± 0.11	
I–II	59.24 ± 11.09		0.38 ± 0.12	
III–IV	60.71 ± 9.05		0.36 ± 0.09	
Surfactant		0.076		0.129
Yes	59.93 ± 9.37		0.37 ± 0.10	
No	65.33 ± 11.28		0.32 ± 0.12	
Apnoea		0.0026 *		0.0039 *
Yes	56.84 ± 9.83		0.40 ± 0.10	
No	65.86 ± 9.45		0.31 ± 0.10	
Inotropic iv		<0.0001 *		<0.0001 *
Yes	54.29 ± 7.04		0.42 ± 0.08	
No	66.68 ± 9.48		0.30 ± 0.10	
Neonatal sepsis		0.106		0.138
Yes	59.26 ± 9.13		0.37 ± 0.09	
No	64.28 ± 10.99		0.33 ± 0.14	
Metabolic acidosis		0.147		0.155
No/mild (pH > 7.2)	66.19 ± 11.33		0.31 ± 0.12	
Moderate (7.0 < pH < 7.2)	60.86 ± 9.37		0.36 ± 0.10	
Severe (pH < 7)	56.75 ± 12.34		0.41 ± 0.12	
Hypocapnia		0.047 *		0.043 *
Yes	65.04 ± 9.07		0.32 ± 0.09	
No	59.05 ± 11.32		0.38 ± 0.12	

IVH: intraventricular haemorrhage; crSaO_2_: cerebral regional oxygen saturation; cFTOE: cerebral fractional tissue oxygen extraction; ^(b)^ data were expressed using descriptive statistics as mean ± standard deviation and absolute (relative) frequencies; * significant level: *p* < 0.05.

**Table 3 jcm-11-03546-t003:** Bivariate correlation analysis between cFTOE and crSaO_2_ with IR, TAM, tcSaO_2_ and tcSaO_2__FiO_2._

	Average crSaO_2_	cFTOE
Variables	Correlation Coefficient ρ [95% CI]	Correlation Coefficient ρ [95% CI]
IR	−0.06 [−0.35, 0.25]	0.04 [−0.26, 0.36]
MAP		
Day 1	0.40 [0.15, 0.61] *	−0.40 [−0.61, −0.14] *
Day 2	0.34 [0.07, 0.56] *	−0.33 [−0.53, −0.08] *
Day 3	0.41 [0.16, 0.63] *	−0.39 [−0.62, −0.15] *
Day 4 ^(a)^	0.44 [0.15, 0.69] *	−0.41 [−0.65, −0.14] *
SatO2		
Day 1	0.10 [−0.15, 0.33]	−0.05 [−0.28, 0.21]
Day 2	0.05 [−0.25, 0.32]	0.01 [−0.26, 0.28]
Day 3	0.32 [0.01, 0.59] *	−0.28 [−0.54, 0.04]
Day 4	0.03 [−0.31, 0.34]	0.0001 [−0.31, 0.31]
SatO_2__FiO_2_		
Day 1 ^(b)^	0.04 [−0.23, 0.31]	−0.02 [−0.29, 0.29]
Day 2	0.17 [−0.11, 0.43]	−0.15 [−0.41, 0.14]
Day 3	0.23 [−0.07, 0.50]	−0.23 [−0.47, 0.04]
Day 4 ^(c)^	0.21 [−0.12, 0.48]	−0.21 [−0.48, 0.09]

crSaO_2_: cerebral regional oxygen saturation; cFTOE: cerebral fractional tissue oxygen extraction; ^(a)^ complete case data *n* = 41; ^(b)^ complete case data *n* = 47; ^(c)^ complete case data *n* = 43; 95% CI = 95% the confidence interval for Spearman’s rank correlation coefficient was calculated using the percentile method based on 1000 bootstrapped samples with replacement; * significant level: *p* < 0.05

## Data Availability

All data generated or analysed during this study are included in this published article.

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
