# Peer review of "Association Analysis of the Cerebral Fractional Tissue Oxygen Extraction (cFTOE) and the Cerebral Oxygen Saturation (crSaO2) with Perinatal Factors in Preterm Neonates: A Single Centre Study"

_jcm, 2022, doi:10.3390/jcm11123546_

Round 1

Reviewer 1 Report

Apnea and hemodynamic compromise are already known parameters to influence NIRS parameters. Authors should better mention how did they detached inotrope use and apnea effect from IVH effect statistically.

One can assume that the demonstrated changes may be related to inotrope use or apnea solely in IVH patients.

It is not clearly discussed by the authors that how is this result will benefit the clinical practice. It is much easier, cheaper and reliable to use cranial USG to demonstrate cranial hemorrhage instead of an unreliable and expensive one.

Reviewer 2 Report

I think that two things should be more described: 1)I think that the Author should describe wider NIRS. that is why I recommend to add the citation in the line 48 - 50 to describe in what cases NIRS can be applied: Shi X, Rehrer S, Prajapati P, Stoll ST, Gamber RG, Downey HF. Effect of cranial osteopathic manipulative medicine on cerebral tissue oxygenation. J Am Osteopath Assoc. 2011;111(12):660–666.
I also think that the hypoxia should be also describe deeper. I advise to add more information about hypoxia, maybe Charlene MT Robertson, MD FRCPC, Max Perlman, MBBS FRCP(London) FRCPC, Follow-up of the term infant after hypoxic-ischemic encephalopathy, Paediatrics & Child Health, Volume 11, Issue 5, May/June 2006, Pages 278–282, https://doi.org/10.1093/pch/11.5.278

Reviewer 3 Report

Despite the interesting topic and background, the  the paper suffers from some issues that should be addressed before considering publication.

Results should be better presented, and the statistical model used for assessing correlation sbould be more clearly stated. 

In particular, regression models should be adopted to describe correlation with clinical variables, and further depicted among figures;  the direct or inverse relationship between SctO2/cFTOE with GA and postnatal age should be more easily described.

Discussion must be improved, also including recent papers on similar subjects that have been recently published (ie Mohamed et al 2021, j perinatol).

The sample size should be implemented to generalize the results.

Finally, please double check the paper for minor typos and few scattered grammar errors (ie related with instead of related to; future tenses instead of first/second conditional; etc).

Round 2

Reviewer 1 Report

Congrats! manuscript is much improoved.